# The microeconomic impact of out-of-pocket medical expenditure on the households of cardiovascular disease patients in general and specialized heart hospitals in Ibadan, Nigeria

Folashayo Ikenna Peter Adeniji[1]*, Akanni Olayinka Lawanson[2], Kayode Omoniyi Osungbade[1]

1 Department of Health Policy & Management, College of Medicine, Faculty of Public Health, University of Ibadan, Ibadan, Nigeria, 2 Department of Economics, Faculty of Economics & Management Sciences, University of Ibadan, Ibadan, Nigeria

* folashayoadeniji@yahoo.co.uk

## Abstract

### Background

Cardiovascular diseases (CVDs) present a huge threat to population health and in addition impose severe economic burden on individuals and their households. Despite this, there is no research evidence on the microeconomic impact of CVDs in Nigeria. Therefore, this study estimated the incidence and intensity of catastrophic health expenditures (CHE), poverty headcount due to out-of-pocket (OOP) medical spending and the associated factors among the households of a cohort of CVDs patients who accessed healthcare services in public and specialized heart hospitals in Ibadan, Nigeria.

### Methods

This study adopts a descriptive cross-sectional study design. A standardized data collection questionnaire developed by the Initiative for Cardiovascular Health Research in Developing Countries was adapted to electronically collect data from all the 744 CVDs patients who accessed healthcare services in public and specialized heart hospitals in Ibadan between 4th November 2019 to the 31st January 2020. A sensitivity analysis, using rank-dependent thresholds of CHE which ranged from 5%-40% of household total expenditures was carried out. The international poverty line of $1.90/day recommended by the World Bank was utilized to ascertain poverty headcounts pre-and post OOP payments for healthcare services. Categorical variables like household socio-demographic and clinical characteristics, CHE and poverty headcounts, were presented using percentages and proportions. Unadjusted and adjusted logistic regression models were used to assess the factors associated with CHE and poverty. Data were analyzed using STATA version 15 and estimates were validated at 5% level of significance.

**Data Availability Statement:** All relevant data are within the paper and its Supporting Information files.

**Funding:** This research was supported by the Consortium for Advanced Research Training in Africa (CARTA). CARTA is jointly led by the African Population and Health Research Center and the University of the Witwatersrand and funded by the Carnegie Corporation of New York (Grant No—G-19-57145), Sida (Grant No:54100113), Uppsala Monitoring Centre and the DELTAS Africa Initiative (Grant No: 107768/Z/15/Z). The DELTAS Africa Initiative is an independent funding scheme of the African Academy of Sciences (AAS)'s Alliance for Accelerating Excellence in Science in Africa (AESA) and supported by the New Partnership for Africa's Development Planning and Coordinating Agency (NEPAD Agency) with funding from the Wellcome Trust (UK) and the UK government. The statements made and views expressed are solely the responsibility of the Fellow." The funders had no role in study design, data collection and analysis, decision to publish, or preparation of the manuscript.

**Competing interests:** The authors have declared that no competing interests exist.

**Abbreviations:** CVDs, Cardiovascular diseases; DALYs, Disability Adjusted Life Years; WHO, World Health Organization; LMICs, Low-and middle-income countries; OOP, Out-of-pocket; CHE, Catastrophic health expenditure; NGN, Nigerian Naira.

## Results

Catastrophic OOP payment ranged between 3.9%-54.6% and catastrophic overshoot ranged from 1.8% to 12.6%. Health expenditures doubled poverty headcount among households, from 8.13% to 16.4%. Having tertiary education (AOR: 0.49, CI: 0.26–0.93, $p$ = 0.03) and household size (AOR: 0.40, CI: 0.24–0.67, $p$ = 0.001) were significantly associated with CHE. Being female (AOR: 0.41, CI: 0.18–0.92, $p$ = 0.03), household economic status (AOR: 0.003, CI: 0.0003–0.25, p = <0.001) and having 3–4 household members (AOR: 0.30, CI: 0.15–0.61, $p$ = 0.001) were significantly associated with household poverty status post payment for medical services.

## Conclusion

OOP medical spending due to CVDs imposed enormous strain on household resources and increased the poverty rates among households. Policies and interventions that supports universal health coverage are highly recommended.

## Background

Cardiovascular diseases (CVDs) represent a major cause of morbidity and mortality [1, 2]. Globally, CVDs accounted for roughly 330 million Disability Adjusted Life Years (DALYs) lost in 2013 [3]. Likewise, a report published by the World Health Organization (WHO) revealed that these health conditions are responsible for approximately 17.9 million deaths worldwide every year, which translates to about one-third of global mortality [4, 5]. In addition to the impacts of CVDs on public and population health, heart-related diseases impose substantial economic burdens on individuals and the society [6–9]. In 2010 alone, the global cost of CVDs was staggering, estimated at approximately US$863 billion and this is predicted to rise to almost US$1044 billion by 2030, representing about a 7.3 percent increase during that time [10].

At patient and household levels, CVDs, like every chronic illness, have the potential to cause severe financial hardship due to the increase in the demand for medical services and a consequent rise in household health expenditures, especially in low-and-middle-income countries (LMICs) where the burden of out-of-pocket (OOP) payment remains high [11]. A study conducted to investigate the financial consequence of OOP for medical services revealed that an increasing number of households are incurring catastrophic health expenditures (CHE) and are being pushed into poverty in LMICs [12]. In episodes of health shocks many households de-save and/or resort to borrowing and selling of assets to offset medical bills. In some cases, individuals may not seek healthcare services either due to the inability to pay or as a result of the fear of being impoverished due to medical spending. These oftentimes undermine the goal of equity and equality of access to health care. Similarly, when the costs of treating chronic diseases such as CVDs is disproportionately borne by individuals and households, it results in serious financial catastrophe [13]. This scenario is even worse for households already within the margin of poverty. Although the outcome of impoverishment resulting from CHE can ordinarily be experienced by households, however, it is often aggravated when a member of the household suffers any form of chronic disease. In general, households with lower health stock as a result of ill-health coupled with limited financial protection are often predisposed to impoverishment.

In Nigeria, the health insurance infrastructure remains underdeveloped. Presently, only a small proportion, about 5%, of the population is covered by the financial protection provided under the National Health Insurance Scheme (NHIS) [14]. Meanwhile, individuals covered under this scheme are majorly federal government workers. Other individuals are either able to pay insurance premium to be enrolled in any form of private insurance or remain uninsured. In particular, the economically less viable individuals who are oftentimes found in the informal sector are disproportionately disadvantaged with regards to access to healthcare insurance in the country. Consequently, individuals and households in this group, especially those with chronic health conditions like CVD face increasing risk of incurring CHE which is capable of widening the gap between the rich and the poor and causing a vicious cycle of poverty.

Recently, a number of studies have reported that the burden of CVDs are rising in Nigeria, like in many other sub-Saharan Africa (SSA) countries [15, 16]. But, there has been no disease-specific studies examining the household-level microeconomic impact of OOP payments for the treatment of CVDs in Nigeria. This studies fills this gap by estimating the CHE headcount, the impoverishing effects of medical expenditures as well as the factors associated with CHE and the poverty induced by healthcare spending among patients who accessed healthcare services in general and specialized heart hospitals in Ibadan, Nigeria. Findings in this study will be beneficial for re-iterating the need for achieving equity in healthcare financing in Nigeria, and indeed, in other SSA countries.

## Conceptual framework

The analysis in this paper follows the framework depicted in Fig 1 in a manner that assesses the economic consequence related to OOP expenditures (i.e. both direct and indirect costs) of treating cardiovascular diseases among patients. Without access to a well-developed prepayment system for the consumption of health care, the overall economic burden of accessing healthcare may have cascading/spill-over effects on the economic welfare of individuals and their households. There are usually heterogeneous microeconomic impacts of OOP medical expenditures which is often dependent on the differentials in the income status of individual/households. Economically less viable patients and their families incur CHE and may even be impoverished due to OOP health expenditures. In the literature, there is evidence that for countries without optimal health insurance infrastructure, uninsured patients are faced with the risk of experiencing 2–7 fold higher levels of CHE relative to insured patients [17].

## Methods

### Study design and population

This study adopted a descriptive cross-sectional study design. Participants were patients who were clinically diagnosed to have any of the CVDs and who accessed outpatient and inpatient healthcare services at tertiary and secondary public hospitals, and private specialized cardiac clinics (a total of 5 hospital facilities) within the Ibadan metropolis. Ibadan is a large city in Southwest, Nigeria and the city is renowned as the biggest city in West Africa with respect to land mass. The tertiary hospital included in this study serves as a major referral center for patients seeking healthcare services due to chronic illnesses in Nigeria.

### Sample size, sample selection and study duration

According to Johnston *et. al.* [18], to estimate the sample size for a continuous outcome variable like OOP health payments and assuming that mean (average) cost $\mu$ and standard

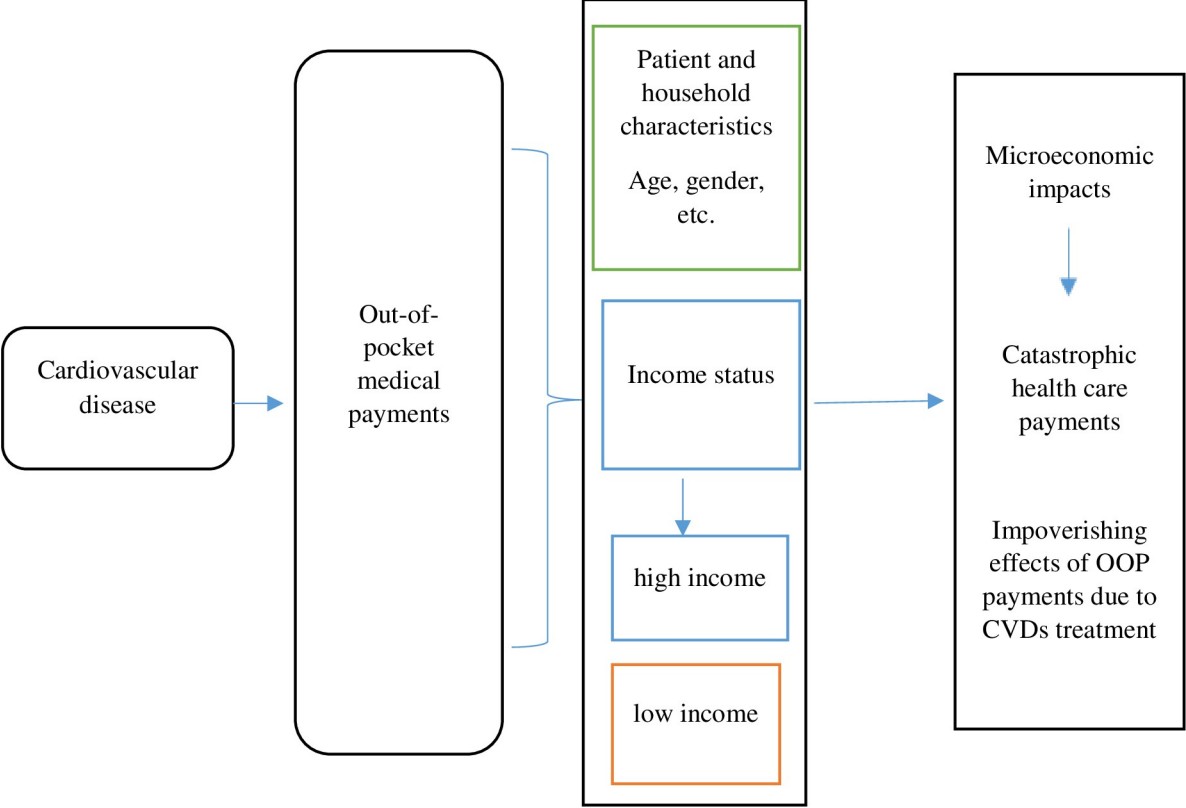

**Fig 1. The microeconomic impact of cardiovascular disease on individuals and households.** Adapted from McIntyre et al. 2006, pp. 860.

deviation $\sigma$ are normally distributed. The width of the precision of a given sample size according can be expressed as:

$$\left(W = 1.96 \times \frac{\sigma}{\sqrt{n}}\right)^2 \tag{1}$$

However, studies with appropriate value for $\sigma$ are rare, therefore to resolve this, Johnston *et. al.* recommended the following formula:

$$\left(\frac{1.96 \times C_v}{V}\right)^2 \tag{2}$$

where $C_v$ denote the coefficient of variation (i.e. the ratio of the standard deviation and the mean cost), $V$ represent the desired level of precision which is 95% confidence interval (CI). The $C_v$ for a 95% CI is 0.50. Thus, the minimum sample size was determined as follows:

$$n = \left(\frac{1.96 \times 0.50}{0.05}\right)^2 = 384 \tag{3}$$

Adjusting the sample size for 10% non-response rate:

$$n_f = \frac{n}{1 - NR} \tag{4}$$

Where $n_f$ denotes non-response and $NR$, non-response rate

$$n_f = \frac{384}{1 - 0.1} = 427 \tag{5}$$

In Ibadan, there is one Federal government owned tertiary hospital facility, two State government owned secondary hospitals with capacities to attend to patients with acute and chronic diseases and three specialized heart hospitals. Data was collected from these hospital facilities and a total sampling of all the 744 CVDs patients who accessed healthcare services from those hospitals between 4[th] November 2019 to the 31[st] January 2020 was carried out. Three hundred and thirty-eight patients were recruited from the tertiary hospital, 368 from the two public secondary health facilities and 38 from all the three specialised heart hospitals

## Data collection and quality control

A standardized data collection questionnaire developed by the Initiative for Cardiovascular Health Research in Developing Countries and used in a previous study [19] was adapted for this study. The instrument elicited information on patients' socio-demographic characteristics, medical/clinical profile, OOP payments for outpatient and inpatient care (for those who had been hospitalized). Also, the respondents were asked to provide the monthly average food and non-food expenditure incurred by their various households. The OOP expenditures includes payment for hospital fees, medicines/drugs, laboratory tests, hospital bed, costs of surgery, emergency room, transportation to and from the hospital and the costs of food during hospital admission. To contextualize and validate the data collection tool, a pre-test was carried out. Data was collected from 43 respondents which represents 10% of the minimum sample size. Two hospital facilities apart from the ones included in the study were used for this purpose. The data collected during the pre-test was not included in the final analysis.

Trained research assistants collected data electronically using the Redcap software [20]. Patients who were attending outpatient clinic were interviewed. Also, patients who had been hospitalized 1 year prior to the period of data collection were asked to provide the health expenditures incurred during their hospital stay. Having co-morbidities could potentially bias cost estimates, therefore, participants were asked to report only the OOP expenditures (both direct and indirect costs) pertaining to CVD treatment.

Quality control was ensured through strict supervision of the data collection process. In some cases, random verification was conducted by reviewing hospital records and through random phone calls to patients and/or their caregivers. The validation of data was carried out on a daily basis by two trained research supervisors. The total expenditures per outpatient visits and the number of monthly outpatient visits were collected. With this, the annual outpatient OOP health expenditure was calculated. This same procedure was followed to annualize OOP payments for inpatient care. These were summed up for patients who assessed both outpatient and inpatient care. All expenditures were collected in the Nigerian Naira (NGN).

## Variables

In this study, the primary outcome variables are CHE headcount and the impoverishing effect of healthcare spending that is related to CVDs treatment. Also, the explanatory variables are respondents' age, gender, educational level, marital status, occupation, existence of comorbidity, whether the respondent was hospitalized in the last 12 months or not, household size as well as household income.

## The microeconomic impact of OOP expenditure

The measure of the microeconomic impact of OOP spending for healthcare service in this study is rather a narrow one as it only considered the CHE headcount and the impoverishment induced by medical expenditures among a cohort of CVDs patients. Specifically, the analysis of the incidence of CHE and the impoverishing effects of healthcare expenditures were carried out at household level following the procedure adopted in previous studies [21–27].

## The measure of CHE

The concept of CHE has been described as the OOP healthcare expenditure over a predetermined proportion, (z%), of household income/resources which is capable of predisposing them to financial distress [28, 29]. However, there is no consensus regarding a specific threshold over which medical spending can be adjudged to be catastrophic given that households have varying income levels and applying a blanket threshold may be less plausible. Instead, studies have used different thresholds ranging from 5% to 40% of household total income (or total non-food expenditure in the case of 40% threshold), a form of sensitivity analysis, to ascertain the incidence of CHE [30]. This study favored the "rank-dependent threshold" approach for determining CHE proposed by Ataguba [21, 31]. This methodology adjusts for vertical equity as well as the diminishing marginal rate of income when examining the catastrophic impact of OOP medical expenditures on household resources [21]. The intuition behind Ataguba's approach is that for low income households, spending a relatively small proportion of their resources to purchase healthcare services may be catastrophic while richer households will need to spend a much higher proportion of their income to experience financial catastrophe. Following Ataguba's work, a rank-dependent threshold, $z^!(\%)$, is a function of a parameter of aversion to inequality, $\gamma$, households' income percentile, $p$, and the initial threshold, $z\%$. This is expressed as follows:

$$z^!(\%) = f(p:\gamma) * z\%, \forall f(p:\gamma) = \gamma(1-p)^{(\gamma-1)}, \gamma \in (0,1) \qquad (6)$$

In this study, $\gamma = 0.8$ and $z\% = 5\%, 10\%, 20\%, 25\%$ of total household expenditure and 40% of total household non-food expenditure or capacity to pay, respectively. Parameter $\gamma$ is less than 1 if households in the lower income strata face lower CHE thresholds (which is our objective in this paper) relative to richer households and vice versa. Aversion to inequality was first popularized by Donaldson and Weymark [32], and has been used by Ataguba to implement the rank-dependent threshold, setting it to equal to 0.8 [21, 33]. In computing the CHE incidence, household expenditure was used as a proxy for household income because the former tends to be more accurately reported and is often slightly impacted by short-run fluctuations relative to household income. Also, individuals tend to be disposed to revealing their expenditures compared to their willingness to disclosing their income, especially in developing countries. As such, household consumption expenditures better reflects the living standard of households [24, 34]. Hence, to ascertain the CHE headcount, the rank-dependent catastrophic overshoot, $O_i^!$, which captures the intensity of catastrophic OOP medical spending among households, was defined as:

$$O_h^! = \max\left(0, \left({}^{OOP_h}/_{TE_h} \text{ or } TnfE_h\right)\right) - z^!(\%) \qquad (7)$$

where $OOP_h$ denotes OOP medical expenditure, $TE_h$, total household expenditure and $TnfE_h$, total household non-food expenditure (also referred to as capacity to pay). Applying the $z^!(\%)$ to household income and denoting whether a household incurred CHE or not by $E_h^!$ (i.e. $E_h^! = 1$ if household incurred CHE and 0, if otherwise), rank-dependent CHE headcount, $CH^!$ is

given as:

$$CH^! = \frac{1}{N}\left(\sum_{h=1}^{N} E_h^!\right) = \mu_{E^!} \tag{8}$$

where N denotes the sample size and $\mu_{E^!}^!$, the mean of $E^!$. Also, the mean rank-dependent positive gap (MPG), like the rank-dependent catastrophic overshoot, reflects the intensity of CHE and it is defined as:

$$MPG^! = \frac{\sum_{h=1}^{N} O_h^!}{\sum_{h=1}^{N} E_h^!} = \mu_{O^!}/\mu_{E^!} \; for \; O_h^! > 0 \tag{9}$$

### Measure of the impoverishing effect of OOP medical expenditure

This measures the increase in the incidence of poverty due to OOP health spending. To evaluate this, the international poverty line of \$1.90/day recommendation by the World Bank was utilized [35]. Following this, the poverty incidence pre and post OOP payment for medical services among CVDs patients were estimated thus:

$$H^{pre} = \frac{1}{N}\left(\sum_{h=1}^{N} P_h^{pre}\right) = \mu_{ppre} \tag{10}$$

$$H^{post} = \frac{1}{N}\left(\sum_{h=1}^{N} P_h^{post}\right) = \mu_{ppost} \tag{11}$$

where $H^{pre}$ and $H^{post}$ respectively depict the poverty incidence before and after the OOP medical expenditure was considered. Also, the difference in both estimates represents the impact of OOP health spending on poverty headcount [21, 33].

### Analysis

Categorical variables like household socio-demographic characteristics, CHE and poverty headcount were presented using percentages and proportions. Logistic regression models were used to assess the factors associated with CHE and poverty. The variables related to the experience of CHE and household poverty status post payment for medical services due to CVDs treatment were examined. Data were analyzed using STATA version 15 and estimates were validated at 5% level of significance.

### Ethics approval and consent to participate

Voluntary informed and written consent was obtained from individual respondents before the commencement of data collection. Ethical approval was obtained from the University of Ibadan/University College Hospital ethics review committee (NHREC/05/01/2008a). Approval was also obtained from respective hospital facilities.

### Results

The socio-demographic characteristics of CVDs patients are presented in Table 1. Majority of the patients were within the age range of 45–74 years. There were more females (68.55%) relative to the number of males. Also, about 28.36% of the respondents had primary education, 27.28%, tertiary education, 24.73%, secondary education while 19.62% had no formal education. The respondents are mostly married: 515 in total accounting for 69.22% of the patients,

**Table 1. Socio-demographic characteristics of CVDs patients accessing healthcare in public and private hospitals in Ibadan (N = 744).**

| Variables | Frequency | Percent (%) |
|---|---|---|
| **Age Group (Years)** | | |
| <45 | 76 | 10.22 |
| 45–54 | 133 | 17.88 |
| 55–64 | 206 | 27.69 |
| 65–74 | 224 | 30.11 |
| >74 | 105 | 14.11 |
| **Gender** | | |
| Male | 234 | 31.45 |
| Female | 510 | 68.55 |
| **Educational Level** | | |
| None | 146 | 19.62 |
| Primary | 211 | 28.36 |
| Secondary | 184 | 24.73 |
| Tertiary | 203 | 27.28 |
| **Marital Status** | | |
| Single | 15 | 2.02 |
| Divorced/Separated | 14 | 1.88 |
| Widow/Widower | 200 | 26.88 |
| Married | 515 | 69.22 |
| **Occupation** | | |
| Employed (government) | 65 | 8.74 |
| Employed (non-government) | 19 | 2.55 |
| Employed (self) | 265 | 35.62 |
| Unemployed | 100 | 13.44 |
| Retired | 152 | 20.43 |
| Artisan | 128 | 17.2 |
| Disabled/Cannot work | 15 | 2.02 |
| **Household Position** | | |
| Father | 214 | 28.88 |
| Mother | 500 | 67.48 |
| Son | 16 | 2.16 |
| Daughter | 9 | 1.21 |
| Others | 2 | 0.27 |
| **Patients' Household size** | | |
| 1–4 | 503 | 67.61 |
| 5–8 | 212 | 28.49 |
| 9–12 | 24 | 3.23 |
| >12 | 5 | 0.67 |

Note: Others refers to individuals residing in the household but are not part of the nuclear family

while more than one-third (26.88%) of the respondents were those who had lost their spouse. A few 14 (1.88%) were either divorced or separated and only 15 (2.02%) were single. Relating to the occupational distribution of the respondents, 477, representing 64.11% were gainfully employed. Out of which 265 (35.62%) were self-employed, 65 (8.74%) were civil servants employed by government, 19 (2.55%) were employed in private organisation settings, while 128 (17.2%) were artisans. On the other hand, 267 (35.89%) were not gainfully employed, of

which, A total of 152 respondents (20.43%) were retirees, while 100 (13.44%) were unemployed, and 15 (2.02%) were disabled/cannot work. In terms of household position, majority of the respondents were mothers (67.48%) while 214 (28.88%) were fathers. Similarly, 67.61% of the respondents were from households with 1–4 household size while 28.49% were from household size of 5–8 inhabitants. Patients who reported household size between 9–12 inhabitants were 24 (3.23%), while those from household size above 12 inhabitants were the least (0.67%).

The clinical profile of respondents is shown in Table 2. Majority (84.01%) of the patients were diagnosed with hypertensive heart disease while 33 (4.44%) and 29 (3.9%) had dilated cardiomyopathy and ischaemic heart disease, respectively. The CVD with the lowest frequency (0.13%) was cor pulmonale. Of the respondents, 299 (40.19%) reported having other co-morbid conditions. Of the CVDs patients, 60.89% reported visiting the hospital once every month and only 2.15% visited the hospital more than four times on average. One hundred and twenty-eight (17.41%) reported hospitalization in the last one year, with majority of them (81.25%) reporting that they were hospitalized once during that time.

**Table 2. Clinical characteristics of CVDs patients accessing healthcare in public and private hospitals in Ibadan (N = 744).**

| Health Issue | Frequency | Percent (%) |
|---|---|---|
| **Heart-related-Condition** | | |
| Alcoholic cardiomyopathy | 3 | 0.4 |
| Anaemic heart failure | 16 | 2.15 |
| Complete heart block | 6 | 0.81 |
| Congenital heart disease | 5 | 0.67 |
| Cor pulmonale | 1 | 0.13 |
| Dilated cardiomyopathy | 33 | 4.44 |
| Hypertensive heart disease | 625 | 84.01 |
| Ischaemic heart disease | 29 | 3.9 |
| Pericardial valvular heart disease | 7 | 0.94 |
| Peripartum cardiomyopathy | 5 | 0.67 |
| Thyroid disease | 3 | 0.4 |
| Other | 11 | 1.48 |
| **Comorbidity** | | |
| No | 445 | 59.81 |
| Yes | 299 | 40.19 |
| **Frequency of visit to health Facility in a month** | | |
| Once | 453 | 60.89 |
| Twice | 147 | 19.76 |
| Thrice | 89 | 11.96 |
| Four times | 38 | 5.24 |
| > 4 times | 17 | 2.15 |
| **Hospitalized in the last 12months?** | | |
| No | 607 | 82.59 |
| Yes | 128 | 17.41 |
| **No of time hospitalized** | | |
| Once | 104 | 81.25 |
| Twice | 18 | 14.06 |
| Thrice | 4 | 3.13 |
| Five times | 2 | 1.56 |

**Table 3. Incidence and intensity of catastrophic out-of-pocket medical expenditures among CVDs patients accessing healthcare in public and private hospitals in Ibadan.**

| Initial threshold (z%) | 5% | 10% | 15% | 20% | 25% | 40% |
|---|---|---|---|---|---|---|
| Catastrophic headcount ($CH^I$) | 54.6% | 33.4% | 21.4% | 10.0% | 3.9% | 7.8% |
| Catastrophic overshoot ($O_i^I$) | 12.6% | 10.9% | 6.3% | 4.7% | 1.8% | 2.2% |
| Mean positive gap ($MPG^I$) | 23.1% | 32.6% | 29.4% | 47.0% | 46.2% | 28.2% |

Note: Estimates were generated at household level. The households of all the 744 patients were included in the analysis

Table 3 shows the incidence and magnitude of CHE incurred by the households of CVDs patients. At 5% initial threshold, the CHE headcount was 54%. The level of financial catastrophe decreased as the thresholds increased. For the 40 percent threshold where the non-food expenditure was used as the denominator, 7.8% of the households incurred CHE. The catastrophic overshoot ranged from 1.8% at 25% initial threshold to 12.6% at 5% threshold. Similarly, the mean positive gap ranged between 23.1% and 47%.

The computed poverty headcount pre and post deduction of OOP medical expenditures is presented in Table 4. The level of poverty was 8.13% before accounting for the monetary outlays towards accessing treatment due to CVDs. This doubled post payment for healthcare services as a further 61 households were impoverished, and consequently, the level of poverty rose to 16.4%.

The distribution of the factors related to the risk of experiencing CHE is presented in Table 5A and 5B. In the unadjusted logistic regression, households of CVDs patients within the 65–74 years and above 74 years age groups were significantly more likely to incur CHE relative to respondents of ages less than 45 years (OR: 1.46, CI: 0.59–2.43 and OR: 2.27, CI: 0.93–4.65, respectively). Similarly, at 10% threshold, respondents within age groups 55–64 years, 65–74 years and ages above 74 years were twice, thrice and four times more likely to incur CHE (OR: 2.03, CI: 1.04–3.97; OR: 3.55, CI: 1.84–6.83; OR: 4.24, CI: 2.09–8.62) compared with respondents within ages less than 45 years. Also, the level of education of patients is significantly associated with the risk of incurring CHE. Participants who had tertiary education were less likely to experience CHE compared with those who had no formal education at 5% threshold (AOR: 0.49, CI: 0.26–0.93, $p$ = 0.03). Similarly, at 10% threshold, CVDs patients who had secondary and tertiary education were significantly less likely to experience CHE (OR:0.62, CI:0.39–0.99; OR:0.51, CI:0.32–0.81) relative to those who had not been formally educated. This pattern was also observed for the economic status of households. At both 5% (OR:0.22, CI:0.14–0.36, $p$ = <0.01) and 10% (OR: 0.24, CI: 0.13–0.41, $p$ = 0.01) thresholds, the richest

**Table 4. Poverty headcount before and after accounting for out-of-pocket medical expenditures among the households CVDs patients accessing healthcare in public and private hospitals in Ibadan (N = 744).**

| Poverty headcount before accounting for OOP health payments | Frequency | Percent (%) |
|---|---|---|
| Poor | 60 | 8.13 |
| Non-poor | 678 | 91.87 |
| Poverty after accounting for OOP health payments | | |
| poor | 121 | 16.4 |
| non-poor | 617 | 83.6 |

Note: Estimates were generated at household level. The households of all the 744 patients were included in the analysis. Poverty headcount was computed at the World Bank recommended $1.90 per day.

**Table 5. Unadjusted and adjusted logistic regression of the factors associated with catastrophic health expenditure among the households of CVDs patients accessing healthcare in public and private hospitals in Ibadan (N = 744).**

| | 5% | | | | | | 10% | | | | | |
|---|---|---|---|---|---|---|---|---|---|---|---|---|
| | **Unadjusted** | | | **Adjusted** | | | **Unadjusted** | | | **Adjusted** | | |
| | **OR** | **95%CI** | **_p_-Value** | **AOR** | **95%CI** | **_p_-Value** | **OR** | **95%CI** | **_p_-Value** | **AOR** | **95%CI** | **_p_-Value** |
| **a)** | | | | | | | | | | | | |
| **Age Group (Years)** | | | | | | | | | | | | |
| <45 (Ref) | | | | | | | | | | | | |
| 45–54 | 1.05 | 0.59, 1.84 | 0.88 | 0.94 | 0.48, 1.83 | 0.85 | 1.47 | 0.72, 3.03 | 0.29 | 1.17 | 0.54, 2.54 | 0.69 |
| 55–64 | 1.46 | 0.86, 2.48 | 0.16 | 1.07 | 0.55, 2.09 | 0.84 | 2.03 | 1.04, 3.97 | 0.04 | 1.06 | 0.49, 2.25 | 0.89 |
| 65–74 | 1.78 | 1.05, 3.02 | 0.03 | 1.20 | 0.59, 2.43 | 0.61 | 3.55 | 1.84, 6.83 | 0.001 | 1.46 | 0.67, 3.17 | 0.34 |
| >74 | 2.27 | 1.24, 4.15 | 0.01 | 2.08 | 0.93, 4.65 | 0.74 | 4.24 | 2.09, 8.62 | 0.001 | 2.11 | 0.90, 4.92 | 0.09 |
| **Gender** | | | | | | | | | | | | |
| Male (Ref) | | | | | | | | | | | | |
| Female | 0.94 | 0.69, 1.28 | 0.71 | | | | 0.81 | 0.58, 1.12 | 0.20 | | | |
| **Educational Level** | | | | | | | | | | | | |
| None (Ref) | | | | | | | | | | | | |
| Primary | 1.25 | 0.81, 1.93 | 0.32 | 1.01 | 0.62, 1.65 | 0.97 | 1.06 | 0.69, 1.63 | 0.80 | 0.97 | 0.59, 1.57 | 0.899 |
| Secondary | 0.66 | 0.42, 1.02 | 0.06 | 0.58 | 0.34, 0.99 | 0.05 | 0.62 | 0.39, 0.99 | 0.04 | 0.71 | 0.41, 1.22 | 0.215 |
| Tertiary | 0.62 | 0.40, 0.95 | 0.03 | 0.49 | 0.26, 0.93 | 0.03 | 0.51 | 0.32, 0.81 | 0.004 | 0.59 | 0.31, 1.13 | 0.113 |
| **Income** | | | | | | | | | | | | |
| Quintile 1 (Ref) | | | | | | | | | | | | |
| Quintile 2 | 1.17 | 0.72, 1.92 | 0.53 | 1.19 | 0.71, 2.01 | 0.51 | 1.37 | 0.86, 2.18 | 0.18 | 1.49 | 0.90, 2.46 | 0.124 |
| Quintile 3 | 0.63 | 0.39, 1.01 | 0.06 | 0.66 | 0.39, 1.12 | 0.12 | 0.71 | 0.44, 1.14 | 0.15 | 1.04 | 0.62, 1.76 | 0.882 |
| Quintile 4 | 0.66 | 0.41, 1.07 | 0.09 | 0.77 | 0.44, 1.35 | 0.36 | 0.41 | 0.25, 0.68 | 0.001 | 0.65 | 0.36, 1.17 | 0.151 |
| Quintile 5 | 0.22 | 0.14, 0.36 | <0.001 | 0.27 | 0.15, 0.49 | 0.001 | 0.24 | 0.13, 0.41 | 0.001 | 0.43 | 0.22, 0.83 | 0.012 |
| **Marital Status** | | | | | | | | | | | | |
| Single (Ref) | | | | | | | | | | | | |
| Divorced/Separated | 5.50 | 1.06, 28.42 | 0.04 | 4.06 | 0.54, 30.29 | 0.17 | 4.88 | 0.78,30.29 | 0.09 | | | |
| Widow/Widower | 1.63 | 0.56, 4.75 | 0.37 | 0.41 | 0.11, 1.59 | 0.20 | 3.94 | 0.87, 17.96 | 0.08 | | | |
| Married | 1.87 | 0.66, 5.33 | 0.24 | 0.94 | 0.26, 3.45 | 0.09 | 3.06 | 0.68, 13.73 | 0.14 | | | |
| Household Size | | | | | | | | | | | | |
| 1–2 (Ref) | | | | | | | | | | | | |
| 3–4 | 0.75 | 0.52, 1.09 | 0.13 | 0.95 | 0.63, 1.44 | 0.80 | 0.50 | 0.35, 0.72 | 0.001 | 0.59 | 0.39, 0.88 | 0.011 |
| 5 and above | 0.4 | 0.27, 0.59 | 0.001 | 0.66 | 0.41, 1.07 | 0.09 | 0.26 | 0.17, 0.40 | 0.001 | 0.40 | 0.24, 0.67 | 0.001 |
| **b)** | | | | | | | | | | | | |
| **Occupation** | | | | | | | | | | | | |
| Employed (government) (Ref) | | | | | | | | | | | | |
| Employed (non-government) | 0.79 | 0.27, 2.29 | 0.67 | 0.64 | 0.20, 2.04 | 0.45 | 0.55 | 0.11, 2.73 | 0.47 | 0.47 | 0.09, 2.46 | 0.37 |
| Employed (self) | 1.56 | 0.90, 2.69 | 0.11 | 0.90 | 0.45, 1.80 | 0.77 | 2.83 | 1.44, 5.54 | 0.003 | 1.50 | 0.67, 3.34 | 0.32 |
| Unemployed | 1.99 | 1.06, 3.76 | 0.03 | 1.13 | 0.49, 2.59 | 0.78 | 3.12 | 1.48, 6.57 | 0.003 | 1.37 | 0.56, 3.36 | 0.50 |
| Retired | 1.91 | 1.06, 3.45 | 0.03 | 1.10 | 0.53, 2.26 | 0.80 | 3.03 | 1.49, 6.13 | 0.002 | 1.50 | 0.66, 3.43 | 0.33 |
| Artisan | 1.18 | 0.65, 2.16 | 0.58 | 0.58 | 0.27, 1.24 | 0.16 | 1.03 | 0.48, 2.22 | 0.94 | 0.43 | 0.17, 1.07 | 0.07 |
| Disabled/Cannot work | 0.93 | 0.29, 2.99 | 0.90 | 0.40 | 0.10, 1.48 | 0.17 | 1.20 | 0.29, 4.99 | 0.80 | 0.34 | 0.07, 1.58 | 0.17 |
| **Comorbidity** | | | | | | | | | | | | |
| No (Ref) | | | | | | | | | | | | |
| Yes | 0.99 | 0.74, 1.34 | 0.96 | | | | 1.01 | 0.74, 1.39 | 0.93 | | | |
| Hospitalized | | | | | | | | | | | | |
| No (Ref) | | | | | | | | | | | | |

(_Continued_)

**Table 5.** (Continued)

| | 5% | | | | | | 10% | | | | | |
| | Unadjusted | | | Adjusted | | | Unadjusted | | | Adjusted | | |
| | OR | 95%CI | *p*-Value | AOR | 95%CI | *p*-Value | OR | 95%CI | *p*-Value | AOR | 95%CI | *p*-Value |
| Yes | 1.34 | 0.91, 2.0 | 0.14 | | | | 0.86 | 0.57, 1.29 | 0.46 | | | |

Note: Income quintile relates to households' income (quintile 1 for the poorest households and quintile 5 for the richest). Other variables are strictly patients' characteristics.

households (households in quintile 5) were less likely to experience CHE compared with patients from the poorest households (quintile 1). Retired and unemployed CVDs patients were significantly more likely to incur CHE (OR:1.91; CI: 1.06–3.45, *p* = 0.03 and OR:1.99, CI: 1.06–3.76, *p* = 0.03). Patients who reported that they live in households with 5 or more household members were less likely to incur CHE relative to those from households with 1–2 members (OR:0.4, CI:0.27–0.59, *p* = <0.01). This was also significant in the unadjusted and adjusted models at 10% threshold (OR: 0.26, CI: 0.17–0.40, *p* = 0.001, AOR: 0.40, CI: 0.24– 0.67, *p* = 0.001).

The distribution of factors associated with poverty after accounting for out-of-pocket medical expenditures is presented in Table 6A and 6B. Respondents in age groups 55–64 years, 65– 74 years and ages above 74 years were significantly more likely to be poor (OR: 3.49, CI:1.19- 10-22, *p* = 0.02; OR: 5.13, CI: 1.78–14.74, *p* = 0.002; OR: 5.33, CI: 1.77–16.10, *p* = 0.003) and were exposed to a higher risk of impoverishment due to OOP medical payments relative to those below the age of 45 years. Female patients were less likely to be poor compared to males (AOR: 0.41, CI: 0.18–0.92, *p* = 0.03). Respondents who had secondary education (OR: 0.33, CI: 0.18–0.60, *p* = <0.001) and those who had tertiary education (OR: 0.22, CI: 0.11–0.41, *p* = < 0.001) were less likely to be poor relative to CVD patients with no formal education. The economic status of the households of patients is significantly associated with their poverty status. The richest households were less likely to be poor relative to those in the lowest income category (AOR: 0.003, CI: 0.0003–0.25, p = <0.001). Also, households with 3–4 members were less likely to be poor compared with those with 1–2 members (AOR: 0.30, CI: 0.15–0.61, *p* = 0.001).

## Discussion

This study assessed the microeconomic impact of OOP medical expenditures by estimating the incidence and extent of CHE, the poverty headcount pre and post-payment of OOP medical expenditures as well as the associated factors among CVDs patients accessing healthcare services at public and private hospital facilities in Ibadan, Nigeria. Findings revealed that having to pay OOP for medical services predisposes CVDs patients to incurring CHE and also increased the poverty rate among the households of those patients. The incidence of financial catastrophe is expectedly the highest when a 5% threshold is used, as a staggering 54.6% of the households of CVDs patients incurred CHE. Although, the CHE headcount decreased with higher thresholds, the level of financial strain on household resources was enormous with respect to the level of CHE for the other thresholds explored.

Furthermore, the study revealed a high intensity and magnitude of CHE among the study population, with this ranging between 2.2%-12.6% for the catastrophic overshoot and between 23.1%-47.0% for the mean positive gap. Given the enormous costs of treating most chronic diseases like CVDs, the evidence in this study is supported by findings in previous studies

**Table 6. Unadjusted and adjusted logistic regression of the factors associated with poverty after accounting for out-of-pocket medical expenditures among CVDs patients accessing healthcare in public and private hospitals in Ibadan (N = 744).**

| | Unadjusted | | | Adjusted | | |
|---|---|---|---|---|---|---|
| | OR | 95%CI | P-Value | AOR | 95%CI | P-Value |
| **a)** | | | | | | |
| **Age Group (Years)** | | | | | | |
| <45 (Ref) | | | | | | |
| 45–54 | 1.62 | 0.50, 5.29 | 0.42 | 1.05 | 0.18, 6.21 | 0.96 |
| 55–64 | 3.49 | 1.19, 10.22 | 0.02 | 1.96 | 0.39, 9.96 | 0.42 |
| 65–74 | 5.13 | 1.78, 14.74 | 0.002 | 1.17 | 0.22, 6.16 | 0.85 |
| >74 | 5.33 | 1.77, 16.10 | 0.003 | 0.98 | 0.16, 5.92 | 0.98 |
| **Gender** | | | | | | |
| Male (Ref) | | | | | | |
| Female | 1.65 | 1.05, 2.60 | 0.03 | 0.41 | 0.18, 0.92 | 0.03 |
| **Educational Level** | | | | | | |
| None (Ref) | | | | | | |
| Primary | 0.77 | 0.47, 1.26 | 0.3 | 0.84 | 0.37, 1.89 | 0.68 |
| Secondary | 0.33 | 0.18, 0.60 | <0.001 | 0.49 | 0.18, 1.33 | 0.16 |
| Tertiary | 0.22 | 0.11, 0.41 | <0.001 | 0.20 | 0.05, o.74 | 0.016 |
| **Income** | | | | | | |
| Quintile 1 (Ref) | | | | | | |
| Quintile 2 | 0.050 | 0.026, 0.096 | <0.001 | 0.028 | 0.013, 0.06 | <0.001 |
| Quintile 3 | 0.006 | 0.001, 0.027 | <0.001 | 0.004 | 0.001, 0.20 | <0.001 |
| Quintile 4 | 0.007 | 0.0016, 0.03 | <0.001 | 0.007 | 0.001, 0.03 | <0.001 |
| Quintile 5 | 0.003 | 0.0004, 0.024 | <0.001 | 0.003 | 0.0003, 0.25 | <0.001 |
| **Marital Status** | | | | | | |
| Single (Ref) | | | | | | |
| Divorced/Separated | 0.31 | 0.03, 3.38 | 0.34 | | | |
| Widow/Widower | 1.52 | 0.41, 5.60 | 0.53 | | | |
| Married | 0.56 | 0.15, 2.04 | 0.38 | | | |
| Household Size | | | | | | |
| 1–2 (Ref) | | | | | | |
| 3–4 | 4.02 | 2.19, 7.37 | <0.001 | 0.30 | 0.15, 0.61 | 0.001 |
| 5 and above | 9.25 | 3.81, 22.44 | <0.001 | 0.41 | 0.15, 1.14 | 0.088 |
| **b)** | | | | | | |
| **Occupation** | | | | | | |
| Employed (government) (Ref) | | | | | | |
| Employed (non-government) | 1.91 | 0.32, 11.35 | 0.48 | 0.60 | 0.03, 6.21 | 0.55 |
| Employed (self) | 3.13 | 1.08, 9.06 | 0.04 | 0.76 | 0.13, 5.01 | 0.83 |
| Unemployed | 7.62 | 2.55, 22.78 | <0.001 | 2.01 | 0.30, 13.98 | 0.46 |
| Retired | 2.35 | 0.77, 7.16 | 0.13 | 0.64 | 0.12, 4.11 | 0.71 |
| Artisan | 2.04 | 0.65, 6.43 | 0.22 | 0.20 | 0.03, 1.38 | 0.10 |
| Disabled/Cannot work | 2.35 | 0.39,14.19 | 0.35 | 0.76 | 0.028, 9.36 | 0.65 |
| **Comorbidity** | | | | | | |
| No (Ref) | | | | | | |
| Yes | 0.76 | 0.51, 1.14 | 0.19 | | | |
| Hospitalized | | | | | | |
| No (Ref) | | | | | | |

(*Continued*)

**Table 6.** (Continued)

| | Unadjusted | | | Adjusted | | |
|---|---|---|---|---|---|---|
| | OR | 95%CI | P-Value | AOR | 95%CI | P-Value |
| Yes | 0.76 | 0.44, 1.32 | 0.33 | | | |

Note: Income quintile relates to households' income (quintile 1 for the poorest households and quintile 5 for the richest). Other variables are strictly patients' characteristics.

conducted in LMICs where there remains a substantial burden of OOP payments due to underdeveloped of health insurance mechanisms. Tolla *et al.* [36] carried out a study that investigated the CHE induced by OOP health expenditures related to CVD prevention and treatment in Addis Ababa, Ethiopia. The authors found that 27 percent of those who accessed CVD care experienced CHE with the poorest households having 60-fold chance of incurring CHE relative to households with the highest economic status. Another study investigated the incidence of financial catastrophe among patients who received inpatient care due to acute coronary event in selected countries in Asia [37]. The study revealed that the burden of CHE was substantial as 66% level of CHE was reported among patients who incurred OOP medical expenditures and are without any form of financial protection in terms of health insurance. This level of CHE appears to be significantly higher compared to the one obtained in this present study. However, a study implemented in India to investigate the CHE headcount among individuals ailing with acute coronary syndrome, reported a relatively much higher incidence of CHE. The study reported that 84 percent of the patients experienced CHE as a result of healthcare spending for the treatment of an episode of acute coronary event [38]. A study that reported a similar incidence of financial catastrophe was that conducted to ascertain the microeconomic effect of OOP payments among hospitalized CVDs patients in four LMICs, Argentina, China, India and Tanzania [19]. The authors stated that more than half of the study participants experienced CHE. Presumably, the level of CHE generated in the reviewed studies would have been impacted by the varying characteristics of the health systems like the quality and cost of health care in the countries where the studies were conducted. For instance, the quality of health infrastructure and thus the quality of healthcare services is often reflected on how much it costs to access healthcare. Also, the heterogeneous methodologies employed in determining the level of CHE may partly be responsible for the different estimates of financial catastrophe. This view aligns with the conclusions made in previous studies [39–41]. Overall, there is abundant research evidence globally that OOP medical expenditures for the treatment of CVDs are quite large and capable of subjecting individuals and their families to financial distress, especially in LMICs where OOP medical expenditures remains the principal source of healthcare financing.

Another evidence revealed in this study is that OOP expenditures due to CVDs treatment among patients had a huge impact on household resources and predisposed them to impoverishment. Specifically, after accounting for health spending towards accessing healthcare services, the poverty rate of patients' households doubled, from 8.13% to 16.4%. This may not augur well for the realization of the goal of poverty reduction as articulated in the Sustainable Development Goals (SDGs). Previous studies have also reported the impoverishing effects of healthcare expenditures in developing countries. A study implemented to estimate the impoverishing effects of health expenditures due to chronic illnesses among household in India revealed that having to pay OOP doubled the poverty rate within the study population [42]. Likewise, Barasa *et al.* examined the poverty impact of direct and indirect OOP healthcare spending in Kenya. The study found that 619,541 Kenyans are impoverished annually due to

OOP medical outlays [26]. A similar study conducted in Egypt reported that there was a 7.4% increase in poverty rate that is attributable to OOP payments for healthcare services among households [43]. Moreover, many other studies have reported similar findings [44–47].

Furthermore, this study assessed the factors associated with CHE and impoverishment due to OOP medical expenditures among households of CVDs patients. The study found that the age of respondents was associated with the experience of CHE, as older patients were more likely to experience financial catastrophe. This finding is intuitive because the severity of disease conditions may increase with age in some cases which is also usually accompanied with higher OOP health expenditures. Apart from this, as individuals attain some certain ages, depending on the country, they are expected to retire from active participation in the labour force. This oftentimes mean that they are able to earn significantly lesser income compared with when they were in active employment. Usually, retired individuals receive pensions in Nigeria, but this pensions are not paid regularly. This coupled with the earlier point may have been the reason why this study showed estimates that suggests that older patients were disproportionately affected by OOP medical payments in relation to the experience of CHE. In the Barasa *et al.* study, results showed that households where the household head (usually the breadwinner) is unemployed and aged were associated with CHE [26]. Also, findings in Tolla *et al.* indicated that the age of respondents is significantly associated with the magnitude of CHE [36].

Consistent with the ideal scenario, the households of CVDs patients who had tertiary education were less likely to incur financial catastrophe due to OOP medical expenditures relative to those with no formal education. This finding is expected because it appears reasonable that there exists a strong correlation between the literacy level of household heads and the economic status of respective households. The idea is that the higher the level of education attained by the household head the more likely the economic prosperity of that household. Likewise, this study revealed that the economic status of households is significantly associated with the experience of CHE. The richest households are less likely to incur CHE relative to the poorest households. This finding has been supported by earlier studies [19, 21, 39, 43, 47]. Another variable related to the economic status of households is the occupation of the respondents. Retired and unemployed CVDs payments were more likely to experience CHE. It is however counter-intuitive that CVDs patients, who reported larger household size are less likely to incur CHE. Nonetheless, in Nigeria it is common that sometimes rich individuals maintain larger households that includes house helps and other relatives who may not necessarily be biological children of the household head.

Moreover, the pattern shown with regards to the factors associated with CHE was also observed for factors related to whether a household is poor or not. Being a female, more viable economic status and household size were significantly associated with the poverty status of the households of CHE patients. In the adjusted model, the households of patients who were females were less likely to be poor relative to those who were males. Perhaps, it can be argued that most of the households in LMICs are headed by males and are usually economically responsible for the survival of their households. In situations where the breadwinner (in this case, a male household head) becomes ill with a chronic disease and unable to work or earn lower incomes due to sick days, this may lead to unimaginable financial hardships for such households. Whereas, if the woman suffers a chronic illness while the man is able to work and support the family financially, the economic impact of chronic illness may not be as severe as what it would have been in the former scenario.

Overall, findings in this study highlights the need for a well-developed financial protection mechanism and suggests that the government needs to do more to scale-up the efforts to achieving universal healthcare coverage. Policy makers should design strategies aimed at

reducing the over-reliance on OOP spending on health and increase options for prepayments to access quality healthcare services. In doing this, government through the NHIS should seek to attain full coverage of the entire population, including those in the formal and informal sectors. This can be achieved by developing a robust tax-based system as well as a social health financing system while encouraging other forms of private health insurance. Indeed, this will be pivotal to preventing a situation whereby individuals and their households are pushed into poverty as a result of large OOP payments for medical expenditures.

## Study limitation

This study is not without limitations. Although, in the execution of the study, careful attention was devoted to ensuring that the information provided by the respondents were verified through the review of medical records, still it is believed that the OOP medical expenditures elicited from participants, especially in relation to inpatient care, might have been affected by recall bias. Similarly, the level of CHE and poverty headcount might be quite higher than what is reported in this study because only OOP medical expenditures incurred for the treatment of CVDs were included in the calculation of the primary outcome variables (i.e. CHE and poverty headcount) which is the focus of this study. However, when the health expenditures incurred in the treatment of other household members are considered, the financial strain and poverty rates due to OOP medical payments as a whole will be larger. Therefore, the interpretation of the findings of this study should be done with full consideration of the stated limitations.

## Conclusion

This study showed that OOP medical spending for accessing healthcare services due to CVDs imposed enormous strain on household resources and increased the poverty rates among households. Given the rising level of CVDs in Nigeria, as also witnessed in many LMICs, there is a need to increase the efforts to ensure that households are not impoverished because of the need to pay OOP for healthcare services. Policies and interventions that supports universal health coverage are highly recommended.

## Supporting information

**S1 Data.**
(CSV)

## Acknowledgments

The authors acknowledge the time and patience of the study participants. Also, the authors wish to thank the reviewers for their valuable comments during the review process.

## Author Contributions

**Conceptualization:** Folashayo Ikenna Peter Adeniji, Akanni Olayinka Lawanson.

**Formal analysis:** Folashayo Ikenna Peter Adeniji.

**Methodology:** Folashayo Ikenna Peter Adeniji, Akanni Olayinka Lawanson.

**Supervision:** Akanni Olayinka Lawanson, Kayode Omoniyi Osungbade.

**Writing – original draft:** Folashayo Ikenna Peter Adeniji.

**Writing – review & editing:** Folashayo Ikenna Peter Adeniji, Akanni Olayinka Lawanson, Kayode Omoniyi Osungbade.

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
