## [Decision Letter · Decision Letter 0]

10 May 2022

PONE-D-21-32628The microeconomic impact of out-of-pocket medical expenditure on the households of cardiovascular disease patients in general and specialized heart hospitals in Ibadan, NigeriaPLOS ONE

Dear Dr. Adeniji,

Thank you for submitting your manuscript to PLOS ONE. After careful consideration, we feel that it has merit but does not fully meet PLOS ONE’s publication criteria as it currently stands. Therefore, we invite you to submit a revised version of the manuscript that addresses the points raised during the review process.

We look forward to receiving your revised manuscript.

Kind regards,

Hao Xue

Academic Editor

PLOS ONE

**Journal requirements:**

“This research was supported by the Consortium for Advanced Research Training in Africa (CARTA). CARTA is jointly led by the African Population and Health Research Center and the University of the Witwatersrand and funded by the Carnegie Corporation of New York (Grant No--B 8606.R02), Sida (Grant No:54100113), the DELTAS Africa Initiative (Grant No: 107768/Z/15/Z) and Deutscher Akademischer Austauschdienst (DAAD). The DELTAS Africa Initiative is an independent funding scheme of the African Academy of Sciences (AAS)’s Alliance for Accelerating Excellence in Science in Africa (AESA) and supported by the New Partnership for Africa’s Development Planning and Coordinating Agency (NEPAD Agency) with funding from the Wellcome Trust (UK) and the UK government. The statements made and views expressed are solely the responsibility of the Fellow”

“"This research was supported by the Consortium for Advanced Research Training in Africa (CARTA). CARTA is jointly led by the African Population and Health Research Center and the University of the Witwatersrand and funded by the Carnegie Corporation of New York (Grant No--B 8606.R02), Sida (Grant No:54100113), the DELTAS Africa Initiative (Grant No: 107768/Z/15/Z) and Deutscher Akademischer Austauschdienst (DAAD). The DELTAS Africa Initiative is an independent funding scheme of the African Academy of Sciences (AAS)’s Alliance for Accelerating Excellence in Science in Africa (AESA) and supported by the New Partnership for Africa’s Development Planning and Coordinating Agency (NEPAD Agency) with funding from the Wellcome Trust (UK) and the UK government. The statements made and views expressed are solely the responsibility of the Fellow".

NO - Include this sentence at the end of your statement: The funders had no role in study design, data collection and analysis, decision to publish, or preparation of the manuscript.”

5. PLOS requires an ORCID iD for the corresponding author in Editorial Manager on papers submitted after December 6th, 2016. Please ensure that you have an ORCID iD and that it is validated in Editorial Manager. To do this, go to ‘Update my Information’ (in the upper left-hand corner of the main menu), and click on the Fetch/Validate link next to the ORCID field. This will take you to the ORCID site and allow you to create a new iD or authenticate a pre-existing iD in Editorial Manager. Please see the following video for instructions on linking an ORCID iD to your Editorial Manager account: https://www.youtube.com/watch?v=_xcclfuvtxQ.

Reviewers' comments:

Reviewer's Responses to Questions

**Comments to the Author**

1. Is the manuscript technically sound, and do the data support the conclusions?

Reviewer #1: Yes

2. Has the statistical analysis been performed appropriately and rigorously? 

Reviewer #1: Yes

3. Have the authors made all data underlying the findings in their manuscript fully available?

Reviewer #1: Yes

4. Is the manuscript presented in an intelligible fashion and written in standard English?

Reviewer #1: Yes

5. Review Comments to the Author

Reviewer #1: The authors have conducted this research in an intelligible manner and in such a way that it is also reproducible by other researchers. There are few comments highlighted and that are detailed in the attached document. Authors are recommended to engage the services of an English editor to improve the readability of the manuscript - paying close attention to the tenses used.

6. PLOS authors have the option to publish the peer review history of their article (what does this mean?). If published, this will include your full peer review and any attached files.

Reviewer #1: No

---

## [Author Response · Author response to Decision Letter 0]

30 May 2022

May 21, 2022

Plos One

Dear Editor,

Subject: PONE-D-21-32628

The microeconomic impact of out-of-pocket medical expenditure on the households of cardiovascular disease patients in general and specialized heart hospitals in Ibadan, Nigeria

Thank you for your letter/email and the opportunity to revise our manuscript. 

We have carefully considered the very valuable suggestions and comments offered by the editor and reviewers. We can say that those comments have been immensely helpful in improving the quality of our manuscript.

We hereby include the editorial comments as well as the reviewer comments below and provided a point-by-point response to all the comments. We included the section, lines and pages where the edits/revisions have been made.

The manuscript has also been scrutinized for the requirements in the author instructions, as well. All the authors contributed and agreed to the revisions.

Thank you very much

Regards,

Dr. Folashayo Adeniji

On behalf of the authors

Editor Comments:

Response: The authors read the Journal’s guidelines for manuscript preparation and our manuscript was formatted accordingly.

Response: This is noted and will be corrected when re-submitting

“This research was supported by the Consortium for Advanced Research Training in Africa (CARTA). CARTA is jointly led by the African Population and Health Research Center and the University of the Witwatersrand and funded by the Carnegie Corporation of New York (Grant No--B 8606.R02), Sida (Grant No:54100113), the DELTAS Africa Initiative (Grant No: 107768/Z/15/Z) and Deutscher Akademischer Austauschdienst (DAAD). The DELTAS Africa Initiative is an independent funding scheme of the African Academy of Sciences (AAS)’s Alliance for Accelerating Excellence in Science in Africa (AESA) and supported by the New Partnership for Africa’s Development Planning and Coordinating Agency (NEPAD Agency) with funding from the Wellcome Trust (UK) and the UK government. The statements made and views expressed are solely the responsibility of the Fellow”

“"This research was supported by the Consortium for Advanced Research Training in Africa (CARTA). CARTA is jointly led by the African Population and Health Research Center and the University of the Witwatersrand and funded by the Carnegie Corporation of New York (Grant No--B 8606.R02), Sida (Grant No:54100113), the DELTAS Africa Initiative (Grant No: 107768/Z/15/Z) and Deutscher Akademischer Austauschdienst (DAAD). The DELTAS Africa Initiative is an independent funding scheme of the African Academy of Sciences (AAS)’s Alliance for Accelerating Excellence in Science in Africa (AESA) and supported by the New Partnership for Africa’s Development Planning and Coordinating Agency (NEPAD Agency) with funding from the Wellcome Trust (UK) and the UK government. The statements made and views expressed are solely the responsibility of the Fellow".

NO - Include this sentence at the end of your statement: The funders had no role in study design, data collection and analysis, decision to publish, or preparation of the manuscript.”

Response: This is noted. The funding statement has been removed from the manuscript and included in the cover letter.

Response: The data used for this study has been uploaded.

5. PLOS requires an ORCID iD for the corresponding author in Editorial Manager on papers submitted after December 6th, 2016. Please ensure that you have an ORCID iD and that it is validated in Editorial Manager. To do this, go to ‘Update my Information’ (in the upper left-hand corner of the main menu), and click on the Fetch/Validate link next to the ORCID field. This will take you to the ORCID site and allow you to create a new iD or authenticate a pre-existing iD in Editorial Manager. Please see the following video for instructions on linking an ORCID iD to your Editorial Manager account: https://www.youtube.com/watch?v=_xcclfuvtxQ.

Response: The ORCD ID of the corresponding author has been added to the Editorial Manager

Response: The ethics statement has been removed from the acknowledgement and taken to the methods section.

Response: We have looked at the references again and made necessary revisions. Kindly refer to the reference section of the manuscript.

Reviewer #1 Comments

1. Change Vancouver formatting style from (9) to [9]. [ ] is the accepted formatting style for PlosOne.

Response: Square brackets have been used in line with journal requirements.

2. On line 89. Consider changing, “This oftentimes undermine the goal….” To: “These oftentimes undermine the goal….” Or “This oftentimes undermines the goal….”

Response: This has been revised to reflect the suggestion of the reviewer. See line 90.

3. On page 499 In Authors’ contributions: Keep authors initials consistent with author names i.e Please remove FA, OL and KO and replace with KIFA. AOL and KOO. 

Response: This has been revised. See line 508

4. In Acknowledgement section: It is always desirable to acknowledge the reviewers also for their roles in improving the paper

Response: The reviewer has been acknowledged in our acknowledgement section.

5. In the Tables, (clearly specify by adding “Ref” category beside all reference categories e.g <45 (Ref); Male (Ref) etc. 

Response: The reference category has been identified. See Tables 5a-6b. 

6. In Table 1, Under Household Positions, include what categories of people you classify as “others’ under the table as a foot note. And please change it from “Other” to “Others”.

Response: This has been included. See Table 1

7. Ensure that tense throughout the results section is reported in past tense. The tense used for reporting in the results seems to alternate between present and past tenses. For instance, on line 264…. “Relating to the occupational distribution of the respondents, 477, representing 64.11% are gainfully employed” should be rephrased as, “Relating to the occupational distribution of the respondents, 477, representing 64.11% were gainfully employed”

Response: Result section has been revised accordingly to ensure that all findings are consistently reported in past tense. See lines 265-362

8. On Line 141, the assertion that Ibadan is renowned to be the biggest city in West Africa only holds true for land mass, not for population or other parameters. Readers may be confused if this is not clearly stated. Please add, “with respect to land mass” to Line 141

Response: This statement has been revised to reflect the suggestion of the reviewer.

9. On Line 184…direct is repeated twice….” only the OOP expenditures (both direct and direct costs) pertaining” …. ought to be,” …only the OOP expenditures (both direct and indirect costs) pertaining…

Response: This has been corrected accordingly

10. On Line 110, “But, there has been no disease-specific studies examining the household-level microeconomic impact of OOP payments for the treatment of CVDs” is a pretty strong assertion to make. I believe what authors mean to say is that this kind of study has not been evaluated in detail yet in SSAs or in developing countries such as Nigeria before. I suggest revising that statement to read, “But, there has been no disease-specific studies examining the household-level microeconomic impact of OOP payments for the treatment of CVDs in Nigeria”.

Response: As suggested by the reviewer, “in Nigeria” has been included to show that the assertion relates to Nigeria.

Overall, the manuscript was read by an expert editor and corrections were made were needed.

The author wish to thank the Editor and the reviewers for their valuable comments.

---

## [Decision Letter · Decision Letter 1]

4 Jul 2022

The microeconomic impact of out-of-pocket medical expenditure on the households of cardiovascular disease patients in general and specialized heart hospitals in Ibadan, Nigeria

PONE-D-21-32628R1

Dear Dr. Adeniji,

We’re pleased to inform you that your manuscript has been judged scientifically suitable for publication and will be formally accepted for publication once it meets all outstanding technical requirements.

Kind regards,

Hao Xue

Academic Editor

PLOS ONE

Reviewers' comments:

Reviewer's Responses to Questions

**Comments to the Author**

1. If the authors have adequately addressed your comments raised in a previous round of review and you feel that this manuscript is now acceptable for publication, you may indicate that here to bypass the “Comments to the Author” section, enter your conflict of interest statement in the “Confidential to Editor” section, and submit your "Accept" recommendation.

Reviewer #1: All comments have been addressed

2. Is the manuscript technically sound, and do the data support the conclusions?

Reviewer #1: Yes

3. Has the statistical analysis been performed appropriately and rigorously? 

Reviewer #1: Yes

4. Have the authors made all data underlying the findings in their manuscript fully available?

Reviewer #1: Yes

5. Is the manuscript presented in an intelligible fashion and written in standard English?

Reviewer #1: Yes

6. Review Comments to the Author

Reviewer #1: I do not have any other concerns about dual publication, research ethics or publication ethics. The authors have done well to address all issues raised.

7. PLOS authors have the option to publish the peer review history of their article (what does this mean?). If published, this will include your full peer review and any attached files.

Reviewer #1: No

---

## [Editor Report · Acceptance letter]

8 Jul 2022

PONE-D-21-32628R1 

The microeconomic impact of out-of-pocket medical expenditure on the households of cardiovascular disease patients in general and specialized heart hospitals in Ibadan, Nigeria 

Dear Dr. Adeniji:

I'm pleased to inform you that your manuscript has been deemed suitable for publication in PLOS ONE. Congratulations! Your manuscript is now with our production department. 

Kind regards, 

on behalf of

Dr. Hao Xue 

Academic Editor

PLOS ONE